# A Noble Extract of *Pseudomonas* sp. M20A4R8 Efficiently Controlling the Influenza Virus-Induced Cell Death

**DOI:** 10.3390/microorganisms12040677

**Published:** 2024-03-28

**Authors:** Su-Bin Jung, Grace Choi, Hyo-Jin Kim, Kyeong-Seo Moon, Gun Lee, Kyeong-Hak Na, Yong Min Kwon, Jimin Moon, Mi Yeong Shin, Jae-Yeong Yu, Yeong-Bin Baek, Jun-Gyu Park, Sang-Ik Park

**Affiliations:** 1Department of Veterinary Pathology, College of Veterinary Medicine and BK21 FOUR Program, Chonnam National University, Gwangju 61186, Republic of Korea; jsbin810@gmail.com (S.-B.J.); ksmoon0409@gmail.com (K.-S.M.); udlrjs77@naver.com (G.L.); ksw06142@naver.com (K.-H.N.); 2Department of Biological Application and Technology, National Marine Biodiversity Institute of Korea, Seocheon 33662, Republic of Korea; gchoi@mabik.re.kr (G.C.); jichi9@mabik.re.kr (Y.M.K.); 3Department of Veterinary Pathology, College of Veterinary Medicine, Chonnam National University, Gwangju 61186, Republic of Korea; gywlsl0420@gmail.com (H.-J.K.); ybbaek@jnu.ac.kr (Y.-B.B.); 4College of Pharmacy, Yeungnam University, Gyeongsan 38541, Republic of Korea; hyp1112@yu.ac.kr; 5Department of Health Research, Jeollanam-do Institute of Health and Environment, Muan 58568, Republic of Korea; bearya@korea.kr (M.Y.S.); brianna0936@korea.kr (J.-Y.Y.); 6Department of Veterinary Zoonotic Diseases, College of Veterinary Medicine, Chonnam National University, Gwangju 61186, Republic of Korea

**Keywords:** *Pseudomonas*, influenza virus, broad-spectrum therapeutics, marine bacterium, cell death

## Abstract

Epidemic diseases that arise from infectious RNA viruses, particularly influenza viruses, pose a constant threat to the global economy and public health. Viral evolution has undermined the efficacy of acquired immunity from vaccines and the antiviral effects of FDA-approved drugs. As such, there is an urgent need to develop new antiviral lead agents. Natural compounds, owing to their historical validation of application and safety, have become a promising solution. In this light, a novel marine bacterium, *Pseudomonas* sp. M20A4R8, has been found to exhibit significant antiviral activity [half maximal inhibitory concentration (IC_50_) = 1.3 µg/mL, selectivity index (SI) = 919.4] against influenza virus A/Puerto Rico/8/34, surpassing the activity of chloroquine. The antiviral response via M20A4R8 extract was induced during post-entry stages of the influenza virus, indicating suitability for post-application after the establishment of viral infection. Furthermore, post-treatment with M20A4R8 extract protected the host from virus-induced apoptosis, suggesting its potential use in acute respiratory disease complexes resulting from immune effectors’ overstimulation and autophagy-mediated self-apoptosis. The extract demonstrated an outstanding therapeutic index against influenza virus A/Wisconsin/15/2009 (IC_50_ = 8.1 µg/mL, SI = 146.2) and B/Florida/78/2015 Victoria lineage (IC_50_ = 3.5 µg/mL, SI = 343.8), indicating a broad anti-influenza virus activity with guaranteed safety and effectiveness. This study provides a new perspective on mechanisms for preventing a broad spectrum of viral infections through antiviral agents from novel and natural origins. Future studies on a single or combined compound from the extract hold promise, encouraging its use in preclinical challenge tests with various influenza virus strains.

## 1. Introduction

Pathogenic microorganisms, particularly infectious viral diseases, continue to threaten human society worldwide [1,2,3]. The majority of these diseases are caused by RNA viruses, including influenza A virus (IAV), which have resulted in catastrophic damage to the national economy and public health [4,5]. The high genetic diversity of RNA viruses accelerates evolutionary adaptation, leading to the acquisition of drug resistance against previously developed vaccines and drugs [6]. Therefore, exploring novel antiviral agents sourced from natural substances has been a critical focus of research, as numerous studies in history have supported their safety and effectiveness [7].

Natural compounds from marine microbes have recently garnered significant attention in drug development due to their anti-inflammatory, antitumor, antimicrobial, antiviral, antimalarial, and antioxidant activities [7,8,9,10,11,12]. Every year, a large number of natural products with antiviral potential are discovered, some of which have even advanced to clinical trials and commercial use [9,13]. Alkaloids, peptides, polyketides, and terpenoids with diverse biological functions have been found in the secondary metabolites of marine bacteria. For instance, cyanobacterial-derived cyanovirin N has been evaluated as an anti-HIV agent in preclinical studies [14]. However, natural compounds of marine microbes have been minimally studied in veterinary science due to a lack of understanding of their mechanism of action in preclinical research.

The identification of antiviral agents derived from marine microbes has emerged as a promising approach to controlling a broad spectrum of animal viral infections. Prominently, an extract of *Mameliella* sp. has been found to exhibit significant protection from IAV infection, presenting a potential broad anti-influenza virus activity [8]. A comprehensive analysis of marine microbes has led to the identification of a novel bacterium, *Pseudomonas* sp. M20A4R8, which exhibits potent antiviral activity.

During the process of molecular replication of IAV, viral proteins such as nucleoprotein (NP), non-structural protein 1 (NS1), matrix protein (M1), and matrix protein 2 (M2) have been found to modulate the apoptotic responses of the host by interacting with cellular factors [15]. Notably, the apoptotic effect of IAV NP contributes significantly to cell death by inhibiting the human antiapoptotic protein Clusterin, which leads to the translocation of Bax into the mitochondria [16]. This phenomenon results in the increased permeability of the mitochondrial membrane, thereby promoting the intrinsic apoptosis pathway and ultimately facilitating IAV replication.

In addition to apoptosis, autophagy is another host cell response interconnected with various viral infections. While autophagy is emerging as a central component of antimicrobial host defense against diverse infections, pathogens have evolved to evade, subvert, or exploit autophagy, including IAV and hepatitis C virus (HCV) [17,18,19]. Notably, the virus utilizes autophagy machinery to deliver the incoming viral RNA to the translation apparatus and the machinery can also be used during virus entry and replication [19]. In other words, autophagy may have a proviral role, contributing to efficient viral growth.

Although apoptosis and autophagy exhibit antiviral responses, evolving viruses can evade the defense system and exploit host proteins as a proviral function. In this regard, IAV induces apoptosis of neutrophils and macrophages, thereby escaping clearance by immune cells. Furthermore, uncontrolled viral propagation increases innate immune sensing of IAV by multiple host cell pattern recognition receptors (PRRs), producing various pro-inflammatory cytokines. In IAV- and SARS-CoV-2-infected patients, severe lung injuries induced by cytokine storms such as acute respiratory distress syndrome (ARDS) contribute most to the poor prognosis and fatality [20,21,22].

This study aims to investigate the antiviral properties of a novel strain of *Pseudomonas* and elucidate its therapeutic potential, which could provide new insights into preventing a broad spectrum of viral infections. After characterizing the single or combined compounds within the extract and explaining the anti-apoptotic mechanism, the therapeutic use of the M20A4R8 strain could be warranted for preclinical treatment to develop a broad-spectrum antiviral drug. Future studies on the single or combined compound from the extract will be promising, encouraging its use in preclinical challenge tests with various influenza virus strains.

## 2. Materials and Methods

### 2.1. Isolation of the Bacterial Strain and Culture Conditions

Bacteria, including *Pseudomonas*, *Pseudoalteromonas*, Paraglaciecola, Alteromonas, Marinomonas, and *Kineobactrum* species, were procured from the coastal seawater collected in Dangjin (Chungnam Province, Republic of Korea) on 16 March 2020. For strain isolation, small sections of *R. crispus* roots were homogenized, diluted with sterile 0.85% saline, and spread on 1/10 diluted ZoBell medium (0.5 g peptone, 0.1 g yeast extract, and 0.001 g FePO_4_ per liter of 20% distilled water and 80% aged seawater). The inoculated plates were then incubated at 25 °C for 5 days, followed by streaking of individual colonies based on morphological differences on marine agar 2216 (Difco, Franklin Lakes, NJ, USA). After primary isolation and purification, the strain was routinely cultured on MA at 25 °C and preserved with 20% (*v*/*v*) glycerol at −80 °C. The bacterial isolates were deposited under the numbers MI0000628, MI00006266, MI00006269, MI00006270, MI00006279, and MI00006281, respectively, at the Microbial Marine BioBank (MMBB) of the National Marine Biodiversity Institute of Korea (MABIK).

### 2.2. Phylogenetic Analysis of 16S rRNA Gene

Genomic DNA was extracted using an Exgene DNA extraction kit (Gene All, Seoul, Republic of Korea) as per the manufacturer’s instructions for amplifying the 16S rRNA gene with the bacteria-specific universal primers 27F and 1492R [23]. The amplified partial 16S rRNA gene sequences were performed with an Applied Biosystems automated sequencer (ABI 3730XL) at Macrogen Co., Ltd. (Seoul, Republic of Korea) and assembled by Geneious program v9.0.5 to obtain a nearly full-length 16S rRNA gene sequence. Identification of the phylogenetic position of the M20A4R8 strain was searched using the EzBioCloud server (https://www.ezbiocloud.net/identify, accessed on 30 March 2020) [24], and the 16S rRNA gene sequence (1494 nucleotides) was compared with validly published species in the server. The 16S rRNA gene sequence has been deposited into GenBank under the accession OR481699. Phylogenetic trees based on 1419 unambiguously aligned sequences were reconstructed using the neighbor-joining (NJ), maximum-likelihood (ML), and maximum-parsimony (MP) algorithms [25,26,27] in the MEGA X version 11.0 [28]. To evaluate the robustness of the tree topologies, the 1000 bootstrap resampled datasets [29] were performed in the three algorithms. The M20A4R8 strain and test samples from other Gammaproteobacterial classes were also compared via 16S rRNA gene sequences. Phylogenetic trees were performed in MEGA the 1000 bootstrap resampled datasets were performed in the three algorithms as above.

### 2.3. Preparation of the Bacterial Extracts

Bacterial extracts were prepared using a slight modification of the method from Choi et al. [30]. Bacteria, including *Pseudoalteromonas*, *Alteromonas*, *Paraglaciecola*, *Pseudomonas*, and *Kineobactrum species*, were first cultured in 2.5 L Erlenmeyer flasks each containing 1 L of marine broth (total 5 L) under 60 µmol m^−2^ s^−1^ LED light at 25 °C. Subsequently, 20 L of culture using a panel or column-type photobioreactor was inoculated with the initial inoculum at a concentration of 10^4^ CFU mL^−1^ in the same culture condition for 10–20 days with shaking at 150 rpm. At the end of the culture period, the culture broth was extracted twice with the same volume of ethyl acetate (EtOAc). The EtOAc soluble component of bacterial culture broth was combined and dried using a vacuum evaporator. A crude total of 150 mg of bacterial extract was obtained from each species. For the antiviral evaluation, the extracts were dissolved in dimethyl sulfoxide (DMSO) (Thermo Fisher Scientific, Waltham, MA, USA).

### 2.4. Cells and Virus

The influenza virus A/Puerto Rico/8/34 (H1N1) (A/PR8), A/Wisconsin/15/2009 (H3N2) (A/Wisconsin) strain, and B/Florida/78/2015 Victoria lineage (B/Florida) were amplified in MDCK cells. The viruses were cultured for 2–3 days in a culture medium supplemented with 1 µg/mL of TPCK-treated trypsin. After collecting centrifuged supernatant, the titer was measured using a cell culture immunofluorescence (CCIF) assay with a monoclonal antibody (Mab) specific for IAV nucleoprotein (NP) protein (Abcam, Cambridge, UK) and expressed as fluorescence focus units per milliliter (FFU/mL) [31].

### 2.5. Cytotoxicity Assessment

A water-soluble tetrazolium salt (WST) assay was used to determine the viability of the cells using the CellVia cell viability assay kit (AbFrontier, Seoul, Republic of Korea) as per the product guide protocol. MDCK cells were seeded in 96-well microplates and treated with each *Pseudomonas* sp. the next day. M20A4R8 extract was used at the following non-cytotoxic concentrations in triplicate: 1000, 500, 250, 100, 50, 25, 10, 5, 2, 1, 0.5, 0.1, and 0.01 µg/mL. After 48 h of incubation at 37 °C, the treated cells were washed with DPBS (Lonza, Basel, Switzerland) and 50 µL of CellVia solution diluted 10 times to DMEM (Welgene, Namcheon, Republic of Korea) in each well. For 30 min, the microplate was incubated at 37 °C. The wells’ optical density (OD) was measured at 450 nm, and the dose–response curve was plotted using GraphPad Prism 9.5.1 (GraphPad Software Inc., San Diego, CA, USA). Extract cytotoxicity was assessed using the following formula: cytotoxicity (%) = (OD_extract_ − OD_Blank_)/(OD_mock_ − OD_Blank_) × 100. The entire process was performed using three independent replicates, with each replicate containing three wells.

### 2.6. Antiviral Screening Assay

The bacterial extract was applied before and after IAV inoculation at an MOI of 0.1 in MDCK cells to confirm general antiviral activity, as described elsewhere [8,32]. Before virus infection, the cells were incubated with serially diluted *Pseudomonas* sp. M20A4R8 extract (starting from concentrations of 500, 250, 125, 62.5, 31.3, 15.6, 5, 1, and 0.1 µg/mL, diluted with DMEM) at 37 °C for 1 h. Subsequently, the cells were washed with DPBS and infected with IAV (0.1 MOI) at 37 °C for 1 h. Following virus adsorption, the cells were washed and treated with the same concentration of bacterial extract as previously described, and then supplemented with 1 µg/mL of TPCK-treated trypsin (Thermo Fisher Scientific).

### 2.7. Time-of-Addition (Application of Drug at Different Time Points) Antiviral Experiment

A set of time-of-addition antiviral experiments with different time points of drug treatment was carried out as follows: treatment before (pre-treatment), during (co-treatment), or after (post-treatment) viral infection at 1 h intervals as previously reported [8,33]. For pre-treatment, the cell was incubated with extract for 1 h, washed with DMEM, and infected with IAV (0.1 MOI). After 1 h, the cell was cleaned and replaced with DMEM supplemented with 1 µg/mL of TPCK-treated trypsin. For co-treatment, the MDCK cells were treated simultaneously with virus and extracts, incubated at 37 °C for 1 h, washed, and replaced with DMEM supplemented with 1 µg/mL of TPCK-treated trypsin. For post-treatment, the cell was infected with the virus. At 1 h, the cell was washed, and different concentrations of *Pseudomonas* sp. were added. *Pseudomonas* sp. M20A4R8 extract diluent was added to DMEM supplemented with 1 µg/mL of TPCK-treated trypsin. Each group was examined in triplicate in 96-well plates, and the supernatants were collected via extract concentration at 48 h post-infection (hpi) for further investigation.

Concerning the antiviral experiment, the half maximal inhibitory concentration (IC_50_) was measured at 570 nm through MTT (3-(4,5-Dimethylthiazol-2-yl)-2,5-diphenyltetrazolium bromide) (Sigma-Aldrich, St. Louis, MA, USA) as previously reported [34,35]. After 48 hpi, the cells were washed with DPBS and then incubated with 200 µL of fresh 0.4 mg/mL MTT solution dissolved in DPBS (pH 7.4) at 37 °C for 4 h. Subsequently, supernatants were removed and DMSO was added to dissolve formazan crystals. The plate was placed on a shaker for 10 min, and absorbance was measured. The inhibition effect was calculated using the following formula: inhibition effect (%) = (OD_extract_ − OD_virus_ mock)/(OD_mock_ − OD_virus mock_) * 100 [36].

The half-cytotoxic concentration (CC_50_) and IC_50_ of the M20A4R8 extract were calculated after generating a dose–response curve using GraphPad Prism 9.5.1. SI was determined as the ratio of CC_50_ to IC_50_ for each group.

### 2.8. Virus Attachment and Penetration Assays

As previously reported, MDCK cells were seeded in 24 healthy plates to determine the effect on virus attachment and penetration into cells [37]. In the attachment assay, the cells were treated with 50 µg/mL of *Pseudomonas* sp. M20A4R8 extract at 4 °C for 30 min. After removing the extract and washing with DPBS, the cells were infected with the virus at an MOI of 0.1 and kept at 4 °C for 1 h. The unattached virus was removed by washing with DPBS and, after incubation at 37 °C for 20 h, the inhibitory activity of the extracts on virus attachment to the cells was measured via relative genome copies by RT-qPCR. An antibody against HA of influenza A virus was used as a positive drug for comparison. In the penetration assay, pre-chilled cells were exposed to the virus while being maintained at 4 °C for 1 h. The unattached virus was removed by washing three times with cold DPBS, and the 50 µg/mL extract was treated at 37 °C for 10 min. To remove the unpenetrated virus, the cells were washed with Tris buffer at a low pH of 3.0; a medium was added and incubated at 37 °C for 20 h. Likewise, the virus in the supernatant was detected through RT-qPCR to measure relative genome copies, and the anti-penetrate effect of the extract was determined. As a favorable drug for comparison, chloroquine, a well-known commercial drug, was used.

### 2.9. Real-Time Quantitative PCR

The reduction in genome copy numbers was confirmed by RT-qPCR quantification of influenza treated with extracts as previously described [38,39]. RNA was extracted from the supernatant of each group of antiviral tests using the QIAamp Viral RNA Mini kit following the handbook’s instructions (QIAGEN, Hilden, Germany). RT-qPCR was performed through the SensiFAST SYBR Lo-ROX One-Step kit (Meridian Bioscience, Cincinnati, OH, USA) using the following primer pairs: IAV PB1 (Forward, 5′-GGC CCT TCA GTT CAT C-3′, Reverse 5′-GCA GAC TTC AGG AAT GTG-3′). The reaction mixture used consists of 2 µL of RNA template, 10 µL of 2× One-step mix, 1.6 µL of primer (forward + reverse), 0.2 µL of reverse transcriptase, and 0.4 µL of RNase inhibitor. Water was filled up to the final total reaction volume for each sample of 20 µL. The cycling condition was performed as follows: 45 °C for 30 m and 95 °C for 10 m, and then the three-step reaction of 95 °C for 10 s, 60 °C for 10 s, and 72 °C for 15 s was repeated 40 times.

### 2.10. TCID_50_ Assay

To measure infection viral progeny to MDCK cells, a tissue culture infective dose (TCID_50_) was computed using the Reed and Muench method [40]. After washing MDCK cells in a 96-well plate with DMEM, the supernatants of each treatment were re-adapted with 10-fold serial dilution with DMEM supplemented 1 µg/mL TPCK-treated trypsin and incubated at 37 °C with 5% CO_2_. At 5 days post-infection, DMEM was removed, the cells were fixed with 4% PFA buffer (Bio-solution, Seoul, Republic of Korea) for 10 min at room temperature, and the supernatant was discarded and washed with tap water. Control groups were treated with only DMEM without trypsin. For staining, a 5% crystal violet solution (DUKSAN, Busan, Republic of Korea) was used, and the titer of influenza A virus was expressed as a TCID_50_/mL value. For statistical evaluation, three independent experiments were performed, and each replicate contains three wells, producing arithmetic mean ± S.D.

### 2.11. Immunofluorescence Assay (IFA)

IFA was performed to determine the protein of viral progeny using an in situ cell death detection kit (Roche, Basel, Switzerland). MDCK monolayers grown on 8-well chamber glass slides were infected or not with influenza virus A/PR8 at an MOI of 1 with 1 µg/mL TPCK-treated trypsin medium as previously reported [31]. After incubating at 37 °C with a 5% CO_2_ atmosphere for 24 h, the cell was fixed with cold 4% paraformaldehyde for 10 min and permeabilized for 10 min with 0.2% Triton X-100 (Sigma-Aldrich, St. Louis, MA, USA). The cell was blocked at room temperature for 1 h with 0.5% BSA in PBS and incubated at 4 °C overnight with the primary antibody against IAV NP (Abcam, Cambridge, UK) on a shaker. After washing twice with PBS, the secondary antibody, AF488-conjugated goat anti-mouse IgG (diluted 1:200) from Thermo Fisher Scientific, was incubated for 1 h at room temperature in the dark. DAPI (Merck Millipore, Burlington, MS, USA) was used to stain the nucleus, and the cells were observed under an LSM 510 confocal microscope.

For the quantification of the positive signal, IFA images were analyzed as described previously [41]. Briefly, 20 images were taken of each group and converted into a pixel unit, and the data were processed and quantified using Adobe Photoshop CS6 (version 22.4.3). Calculations were based on the intracellular number of positive pixels per image that was present in approximately 20 cells on average. The results were expressed as absolute pixel numbers per cell.

### 2.12. Flow Cytometry of DNA Strand Break

The levels of apoptosis were quantified to determine the impact of treatment with *Pseudomonas* sp. M20A4R8 extract and IAV infection on cell death and viral replication. To the previous report that apoptosis-related endonucleases cause DNA cleavage, dUTP labeling was used to detect 3′hydroxyl-terminal DNA [42,43]. Cells were prepared in a 6-well plate at 90% confluence and incubated for 24 h with or without virus and drug at 37 °C. The cells were detached from the plate by trypsin treatment, and the pellet was harvested by centrifugation at 3000 rpm for 2 min. After washing with PBS, it was fixed with cold 4% paraformaldehyde for 10 min and permeabilized for 10 min with 0.2% Triton X-100. The cell was blocked at room temperature for 1 h with 0.5% BSA (Sigma-Aldrich, St. Louis, MA, USA) in PBS and then incubated at 4 °C overnight with primary antibody and apoptotic markers on a shaker. After washing twice with PBS, the secondary antibody, AF647-conjugated donkey anti-rabbit IgG (diluted 1:200) from Thermo Fisher Scientific (Waltham, MA, USA), was incubated at room temperature for 1 h. The data of each sample were digitized under the same conditions of measurement using an AttuneTM NxT flow cytometer (Invitrogen, Waltham, MA, USA) and AttuneTM NxT software v3.1.2 as previously reported [31].

The gating strategy for the analysis of total MDCK cells involved selecting the appropriate population based on forward (FSC) versus side scatter (SSC), as previously reported [31]. Subsequently, the cells were further gated on FSC-Height (FSC-H) versus FSC-Area (FSC-A) to eliminate all the doublets and generate the singlets gate. Finally, all the subpopulations were analyzed for the positive signal for TUNEL (on forward) and IAV NP (on side). Arithmetic means ± S.D were calculated from three independent experiments, each of which contained three wells.

### 2.13. Statistical Analysis

All data were analyzed using the GraphPad Prism software program version 9.5.1, and the results are presented as mean ± the standard error of the mean (SEM). The statistical analysis was carried out by one-way analysis of variance (ANOVA) with Tukey’s correction for multiple comparisons or independent sample *t*-tests (and nonparametric tests), individually annotated in the figure legend. Statistical significance of data is marked as follows: * *p* < 0.05, ** *p* < 0.01, *** *p* < 0.001, **** *p* < 0.0001.

## 3. Results

### 3.1. DNA-Based Identification and Phylogenetic Analysis

Comparative analysis of the 16S rRNA gene showed that the M20A4R8 strain was closely related to Pseudomonas bohemica IA19T with a similarity of 98.56%. Phylogenetic analysis of the 16S rRNA gene using the NJ, ML, and MP algorithms showed that the M20A4R8 strain formed a monophyletic clade with other members of the genus Pseudomonas (Figure 1A). Consequently, sequence comparisons based on almost complete 16S rRNA gene sequences showed a clear affiliation of the isolate to the genus Pseudomonas.

### 3.2. In Vitro Antiviral Screening for Extract of Pseudomonas sp. M20A4R8 Extract

We investigated the inhibitory efficacy of extracts of the same Gammaproteobacteria class related to the Pseudomonas sp. M20A4R8 strain against influenza A virus (A/PR8 strain) (Figure 1B and Figure 2). Antiviral efficacy was mainly assessed using two main indices: CC_50_ and IC_50_. Briefly, vehicle, chloroquine, an FDA-approved antiviral drug, or one of the bacterial extracts were applied before and after either mock or IAV (A/PR8) inoculation at an MOI of 0.1. Antiviral activity against the A/PR8 strain was defined by an inhibitory effect on cytopathic effect (CPE) using the MTT method with slight modification after 48 h of incubation [34,44]. As a result, IC_50_ and SI of chloroquine were calculated as 1.45 and 20.9, respectively. *Pseudoalteromonas carrageenovora*, *Paraglaciecola aquimarina*, and *Kineobactrum salinum* extracts showed low SI, whereas *Alteromonas marina* and *Marinomonas dokdonensis* extract had low cytoxicity and relatively high SI over 10. In particular, *Pseudomonas* sp. M20A4R8 extracts showed high safety and antiviral effectiveness, showing dramatic SI over 900 (Figure 2).

### 3.3. Antiviral Response of M20A4R8 Extract According to Extract Treatment Time against Influenza Virus

Pseudomonas sp. M20A4R8 extract was applied in three different approaches: pre-treatment (Appendix A), co-treatment (Appendix A), and post-treatment (Figure 3). To determine whether the extract had a preventive effect by affecting the viral receptors in cells, the extract was pretreated 1 h before viral infection (Appendix A). Subsequently, the cells were infected with IAV at an MOI of 0.1, and antiviral evaluation was performed at 48 h after infection. However, the antiviral effect was not observed at any concentration (Appendix A).

To determine whether the extract directly interfered with the attachment or entry of the influenza virus into host cells, the virus and the extract were treated simultaneously with the cells. The viral inoculum (at an MOI of 0.1) was mixed with the extract and absorbed into the MDCK cell for one hour of incubation. After 48 h, the cell pellet and supernatant were analyzed with the experiments above (Appendix A). A serial concentration (from 0.01 to 100 µg/mL) of M20A4R8 extract did not show antiviral activity on CPE and viral genome replication by A/PR8 infection (Appendix A), suggesting that A/PR8 infections were not prevented by the pre- or co-treatment.

Finally, the extract was post-treated 1 h after virus infection to determine whether the extract affected later stages, such as the virus replication or synthesis (Figure 3A). Briefly, MDCK cells infected with the A/PR8 strain (0.1 or 1 MOI) received the extract at various concentrations as above. As a result, the extract showed remarkable inhibition of CPE induced by IAV in a dose-dependent manner, producing a great therapeutic index (IC_50_ = 1.4 µg/mL, SI = 823.6) in comparison with chloroquine (IC_50_ = 1.5 µg/mL, SI = 20.9, a commercial anti-IAV drug) (Figure 3B). In parallel, the IFA results suggested gradual inhibition of viral protein synthesis toward higher concentrations of M20A4R8 extract (Figure 3C). Moreover, viral genome replication and progeny production were significantly suppressed by the post-treatment of M20A4R8 extract (Figure 3D,E).

### 3.4. Protection of Host through M20A4R8 Extract from IAV-Induced Cell Death

To specify the initiation of antiviral response mediated by M20A4R8 extract, virus attachment and penetration assays were performed in MDCK cells infected with the A/PR8 strain (1 MOI) (Appendix A). As a result, MDCK cells pre-incubated with M20A4R8 did not suppress virus binding to a cellular receptor in comparison with the group applied with an inoculum mixture of IAV and an antibody against IAV hemagglutinin (HA) before inoculation on the host cells (Appendix A). Interestingly, the cells post-treated with the extract after the virus binding to the host significantly reduced viral genomes (Appendix A). However, it was considered that the antiviral action would only last for a short time, which was insufficient to curb severe infection, given that the pre- and co-treatment resulted in no inhibition of viral infection during long-time incubation at 48 h. In other words, the potent antiviral activity in Figure 3 was highly expected to be linked to the late stage of viral replication or post-entry stages.

IAV promoted host cell death by induction of the mitochondrial apoptosis pathway. Many studies have found that IAV-induced apoptosis facilitates viral replication by disseminating and killing the host immune effectors [16,45]. Apoptotic cell death was analyzed by flow cytometry to evaluate host cell protection from IAV infection (Figure 4). Apoptotic DNA fragmentations were detected by TUNEL labeling with FITC, and IAV-infected cells were recognized by monoclonal antibody against IAV NP conjugated with Alexa Fluor (AF) 647. M20A4R8 extract did not affect apoptosis, given no significant difference between the mock-infected, vehicle-treated group and the mock-infected, M20A4R8 extract-treated group (Figure 4A,B). The A/PR8-infected, vehicle-treated group showed apoptotic cell death (approximately 12%) by severe viral replication (about 42%) in the host (Figure 4C). However, post-treatment of M20A4R8 extract could counteract virus-induced cell death, dramatically reducing viral replication (approximately 0.2%) (Figure 4D). The antiviral response was evidenced by a significant reduction in apoptosis (Figure 4E) and viral proteins (Figure 4F). Collectively, our data strongly indicated that the bacterial extract efficiently protects the host in parallel with the attenuation of apoptotic cell death.

### 3.5. A Broad Antiviral Activity of M20A4R8 Extract against Different Influenza Virus Strains

To measure in vitro broad-spectrum antiviral effect, antiviral evaluation against multiple strains of influenza virus was performed. As expected, the post-treatment of M20A4R8 extract dramatically exerted antiviral efficacies against influenza virus A/H3N2 (IC_50_ = 8.1 µg/mL, SI = 146.2) and B/Florida (IC_50_ = 3.5 µg/mL, SI = 343.8) at an MOI of 0.1, which were much greater than chloroquine (Figure 5A–D).

Notably, the influenza virus can induce systemic infection in various hosts, causing damage to major organs such as the heart, spleen, kidney, brain, liver, and lymphoid organs [46]. To confirm the antiviral efficacy in the respiratory epithelial cells, A549 cells were infected with the A/PR8 strain (0.1 MOI). As a result, the extract efficiently protected lung epithelial cells from influenza virus A/PR8 (IC_50_ = 1.9 µg/mL, SI = 403.6), much more significantly than chloroquine (Figure 5E,F). These data suggested that M20A4R8 extract has in vitro broad-spectrum antiviral potential, including multiple influenza viruses in various cell lines from different organs.

## 4. Discussion

The emergence and re-emergence of infectious RNA viruses pose a constant threat to public health and global economies. Existing vaccines and antiviral drugs are often rendered ineffective due to the evasion of the host immune response or genetic mutations, leading to the accumulation of pandemic risks. Developing new antiviral lead agents to combat problematic RNA viruses has become an urgent necessity.

In this study, in vitro antiviral screening resulted in the identification of a bacterial extract exhibiting significant antiviral activity against the PR8 strain of influenza A virus, with an IC_50_ of 1.3 µg/mL and an SI of 919.4. The level of activity displayed by this extract is comparable to that of chloroquine, an FDA-approved antiviral drug, and is an encouraging indication of its safety and efficacy, with the potential to be a potent weapon against IAV infection. The bacterial extract, identified as *Pseudomonas* sp. M20A4R8 is a new strain of *Pseudomonas* initially reported in seawater worldwide.

Notably, *Pseudomonas aeruginosa*, a well-known strain of the *Pseudomonas* genus, is an opportunistic pathogen that causes cystic fibrosis, burns wounds, immunodeficiency, and chronic obstructive pulmonary disorder [47,48]. However, antiviral peptides derived from *Pseudomonas chlororaphis* O6 or *Pseudomonas* sp. can effectively inhibit viral replication against Tobacco Mosaic Virus, Oncorhynchus masou virus, and infectious hematopoietic necrosis virus [49]. Therefore, further metabolic and chemical investigations are imperative to identify the specific component(s) or metabolite(s) responsible for the antiviral activity of *Pseudomonas* sp. M20A4R8 extract.

Many antiviral therapeutics have been found to disrupt particular stages of viral replication, such as chloroquine, a drug preventing endosomal acidification, and remdesivir, a drug suppressing RNA-dependent RNA polymerase of SARS-CoV-2 [48,50,51,52]. Previous studies have suggested plausible explanations for the antiviral mechanisms of *Pseudomonas* sp. [53,54]. For instance, *Pseudomonas aeruginosa* (strain PM1) has been reported to exhibit inhibition of viral attachment against tobacco mosaic virus and Sunn-hemp rosette virus [53]. In addition, *Pseudomonas aeruginosa* has been found to enhance acute inflammation in the airway, promoting neutrophil response and antiviral activity against rhinovirus [54]. Time-of-addition experiments have been carried out to investigate the related mechanisms of the antiviral action of *Pseudomonas* sp. M20A4R8 extract. The inhibitory effects were not involved in viral attachment or direct interaction with virus particles according to pre- and co-treatment and attachment and penetration assays. As a result, post-treatment has been found to be the most effective approach, possibly inhibiting genome replication and protein synthesis with a remarkable therapeutic index against influenza virus A/PR8, with an IC_50_ of 1.4 μg/mL and an SI of 823.6. This approach is more adaptable in clinical uses and a more feasible strategy for controlling epidemic diseases than prophylactic therapy.

Influenza virus infections have been found to induce apoptosis mediated by a multifactorial process. The mitochondrial apoptosis pathway is mainly promoted by IAV NP, resulting from the counteraction of cellular anti-apoptotic protein Clusterin [16]. The influenza virus also stimulates host autophagy, activating Bax/caspase-dependent apoptosis [17]. Modulation of virus-induced apoptosis is essential in host protection, preventing the death of immune effectors and secondary bacterial infection to respiratory organs. M20A4R8 extract efficiently suppresses IAV-induced apoptosis and dramatically reduces viral replication. In another way, it is due to a reduced amount of viral protein that could not stimulate PRR-mediated apoptosis. In cases of pulmonary infection caused by pathogenic microorganisms, ARDS is a significant complication with a high percentage of fatality, especially those caused by the influenza virus. This is primarily due to apoptotic cell death and impaired efferocytosis [20,21,22]. The interaction of viral components as a pathogen-associated molecular pattern with PRRs can induce a cytokine storm, resulting in severe lung injury [31].

Moreover, the extract of M20A4R8 has demonstrated remarkable antiviral potential, exhibiting broad-spectrum antiviral activity against different strains of the influenza virus with outstanding therapeutic indexes. It has been found to be effective against influenza virus A/Wisconsin (IC_50_ = 8.1 µg/mL, SI = 146.2) and B/Florida (IC_50_ = 3.5 µg/mL, SI = 343.8) and has been shown to be effective in a lung epithelial cell line (A549) against A/PR8 infection (IC_50_ = 1.9 µg/mL, SI = 403.6), which is significantly higher than that of chloroquine. Although further investigation in primary human respiratory epithelial cells is required in future studies to see the therapeutic feasibility for clinical use, M20A4R8 extract showed great antiviral potential.

Toxicity poses a significant challenge in the therapeutic application of various drugs with antiviral activity [55,56]. Notably, the bacterial extract, M20A4R8, has demonstrated potent inhibition of multiple influenza virus strains at low concentrations, with a relatively high selectivity index compared to FDA-approved chemicals [57]. These findings suggest that M20A4R8 extract may be a safe and effective therapeutic agent. However, further research is necessary to identify the specific compound(s) responsible for modulating apoptosis, using liquid chromatography–tandem mass spectrometry as an additional investigative tool. Moreover, the antiviral evaluation should be conducted through quantitative and qualitative approaches, including assessments of viral transcription, protein synthesis using Western blot and in vitro translation assay, and packaging and assembly. In this regard, future studies on individual or combined compounds from the bacterial extract using significant validation methods will provide further insights into its antiviral properties.

## Figures and Tables

**Figure 1 microorganisms-12-00677-f001:**
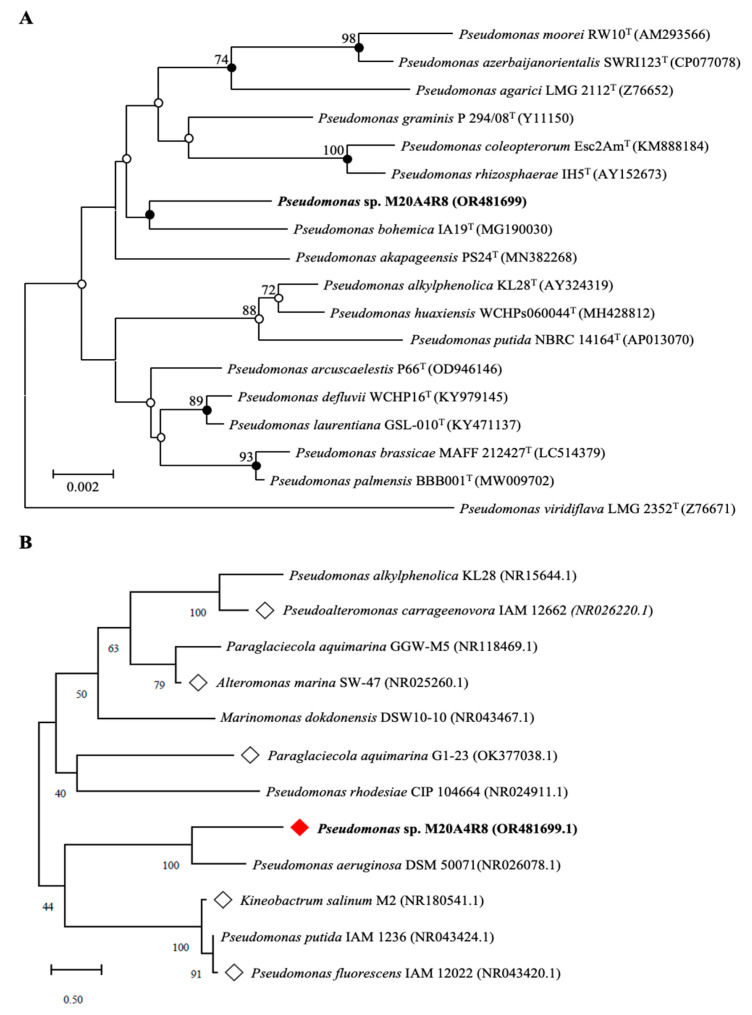
Phylogenetic tree of a *Pseudomonas* sp. M20A4R8. White squares are bacteria whose extracts show significant potential through antiviral screening. A red square is a bacterium whose extract exhibited the most significant antiviral efficacy. (**A**) A neighbor-joining tree based on 16S rRNA gene sequences, showing the phylogenetic relationships of the M20A4R8 strain (highlighted in bold) and closely related taxa with validly published names. GenBank accession numbers are given in parentheses. Bootstrap values above 70% are shown at nodes as percentages out of 1000 replicates. Bar, 0.002 changes per nucleotide position. (**B**) The tree associated with test marine bacteria (rhombic nods) belongs to the same class as *Pseudomonas* sp. M20A4R8. Bootstrap values above 70% are shown at nodes as percentages out of 1000 replicates. Bar, 0.50 change per nucleotide position.

**Figure 2 microorganisms-12-00677-f002:**
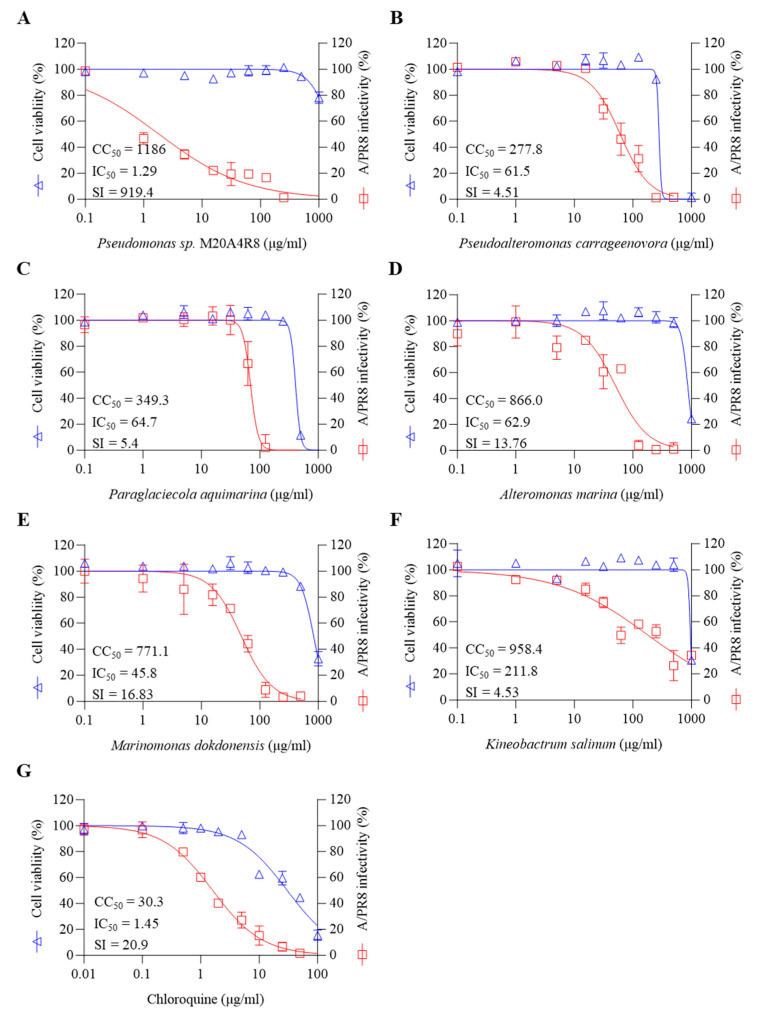
Marine bacteria whose extracts were subjected to preliminary antiviral screening against A/PR8 by pre- and post-treatment in MDCK cells. (**A**) *Pseudomonas* sp. M20A4R8, (**B**) *Pseudoalteromonas carrageenoora*, (**C**) *Paraglaciecola aquimarina*, (**D**) *Altermonas marina*, (**E**) *Marinomonas dokddonensis*, (**F**) *Kineobactrum salinum*, and (**G**) Chloroquine. All data in the graphs are presented as arithmetic means ± S.D. from 3 independent experiments.

**Figure 3 microorganisms-12-00677-f003:**
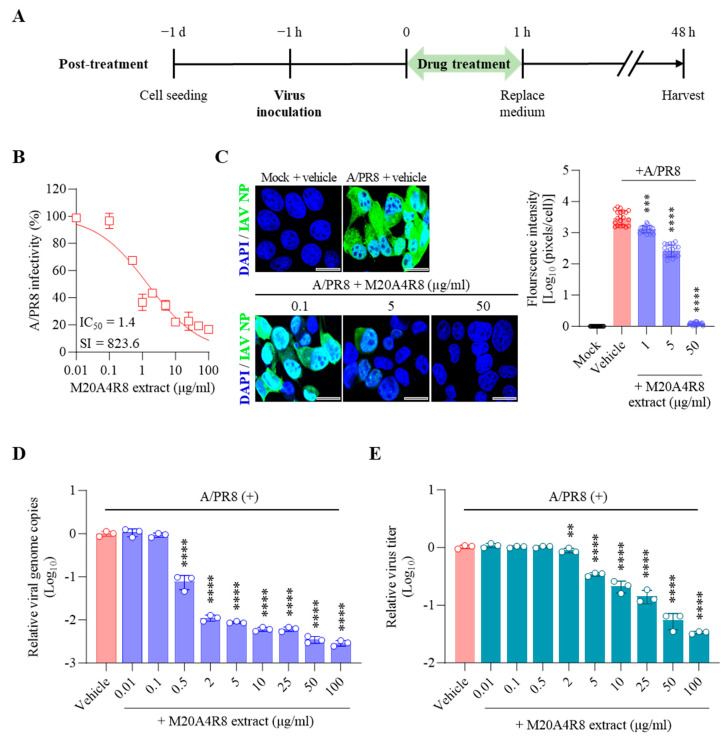
Antiviral effect of M20A4R8 extract post-treatment after infection with influenza virus A/PR8 (0.1 MOI) in MDCK cells. (**A**) Schematic diagram of virus inoculation and extract treatment. (**B**) IC_50_ and SI of M20A4R8 extract measured by CPE-inhibition assay. (**C**) Representative IFA images (on the left) of virus replication under vehicle or extract treatment using an anti-IAV NP antibody (AF488). Cell nuclei were stained with DAPI. The quantification (on the right) of viral protein was produced by mathematical calculation of positive signals per cell. Bar = 20 µm. (**D**) Viral genome copies detected by RT-qPCR. (**E**) TCID_50_ measured progeny virus production. All data in the graphs are presented as arithmetic means ± S.D. from 3 independent experiments. **, *p* < 0.01; ***, *p* < 0.001; ****, *p* < 0.0001.

**Figure 4 microorganisms-12-00677-f004:**
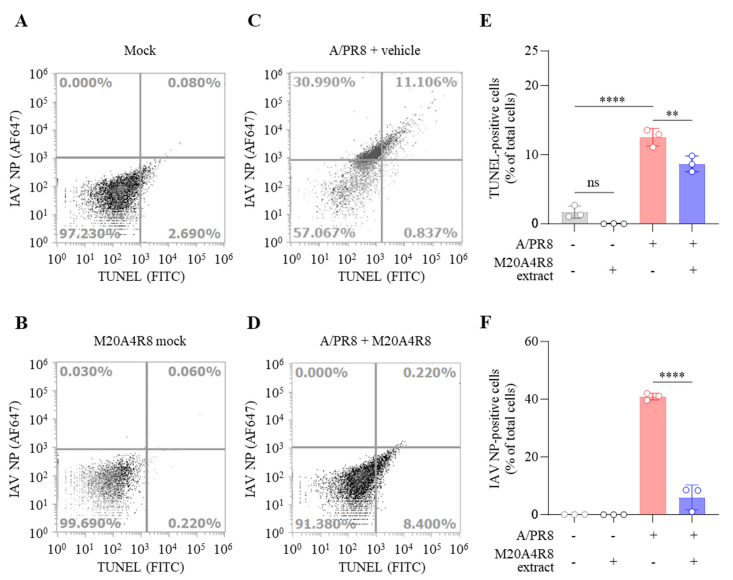
Flow cytometry-measured antiviral evaluation targeting apoptotic reaction of MDCK cells infected with mock or influenza virus A/PR8 (1 MOI) and treated with vehicle or M20A4R8 extract. The apoptosis and viral protein levels were measured by TUNEL assay and immunoreaction targeting IAV NP, respectively. Mock-infected, vehicle-treated group (**A**); mock-infected, M20A4R8 extract-treated group (**B**); A/PR8-infected, vehicle-treated group (**C**); and A/PR8-infected, M20A4R8 extract-treated group (**D**). Graphs summarizing flow cytometry data (**E**,**F**). The fluorescence quantification of TUNEL- (**E**) or IAV NP-positive cells (**F**), which were treated with the vehicle or M20A4R8 extract. All data in the graphs are presented as arithmetic means ± S.D. from 3 independent experiments. **, *p* < 0.01; ****, *p* < 0.0001. ns, not significant, IAV NP, nucleoprotein.

**Figure 5 microorganisms-12-00677-f005:**
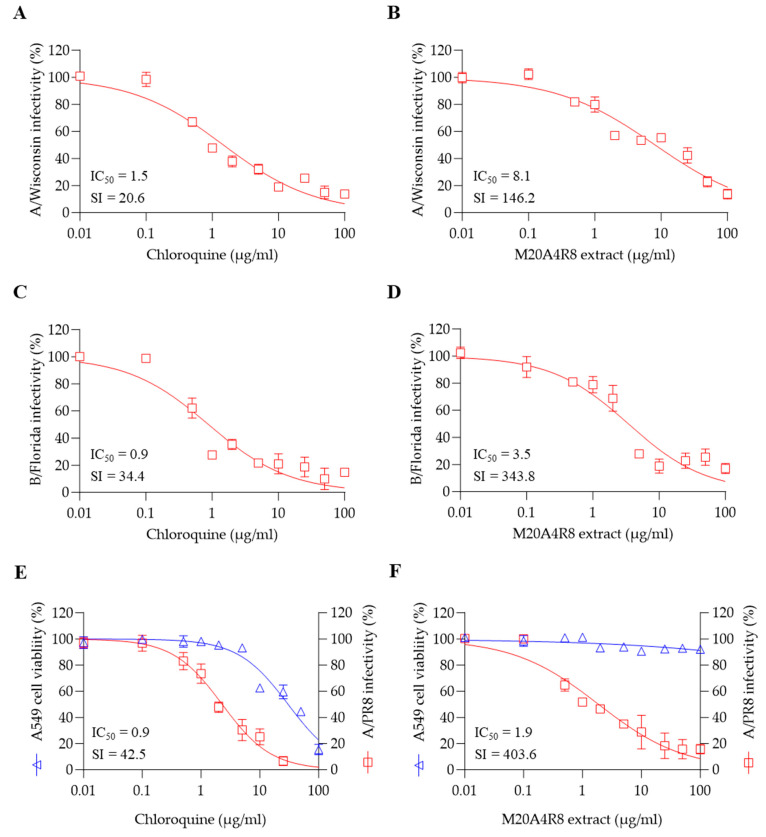
The broad-spectrum antiviral potential of Pseudomonas sp. M20A4R8 extract determined by post-treatment against multiple influenza viruses. Antiviral action of chloroquine (**A**) and M20A4R8 extract (**B**) against the A/Wisconsin strain in MDCK cells. Antiviral action of chloroquine (**C**) and M20A4R8 extract (**D**) against the B/Florida strain in MDCK cells. Antiviral activity with chloroquine (**E**) and M20A4R8 extract (**F**) in A549 cells against the A/PR8 strain. All data in the graphs are presented as arithmetic means ± S.D. from 3 independent experiments.

## Data Availability

Data are contained within the article and Appendix A.

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
