# Peer review of "A Noble Extract of Pseudomonas sp. M20A4R8 Efficiently Controlling the Influenza Virus-Induced Cell Death"

_microorganisms, 2024, doi:10.3390/microorganisms12040677_

Round 1

Reviewer 1 Report

Comments and Suggestions for Authors

A noble extract of Pseudomonas sp. M20A4R8 efficiently controlling the influenza virus-induced cell death.

Jung et al. performed a study showing that a bacterial extract, from marine Pseudomonas sp, reduces Influenza virus replication in cell lines, with a negligeable cytotoxic effect. The authors further show that this bacterial extract apparently protects the cell host infection by reducing apoptosis. The manuscript results are clear and to the point, but I have a series of minor comments.

Minor points

Introduction.

-        Some sentences should in fact be either in the results or even in the discussion part : sentence lines 67-70; sentences lines 71-78; sentences lines 83-90.

-        The authors could detail a bit the interplay between influenza infection and apoptosis. It is not clear like it is written.

Materials and methods

-        Line 218 : coronavirus ? not necessary to mention in this article.

-        Lines 246 and 260 : mention which primary antibody has been used : what is the antigen and give references of these antibodies.

-        Lines 247 and 261 : as well, give the references of the secondary antibodies.

Results

-        Please indicate the cell line used in figures 2, S1, 3, 4, 5A-D.

-        Sentence line 327-328 : The viral inoculum (at an MOI of 0.1) was mixed with the extract and absorbed into the MDCK cell, add “for one hour incubation”.

-        Figure 4A/B : the authors should present the flow cytometry results on mock-infected cells in order to assess that the bacterial extract is not inducing itself apoptosis, at least as supplementary data.

-        Sentences lines 394-404: the conclusion that “M20A4R8 extract has in vitro broad-spectrum antiviral potential, including multiple influenza viruses in various cell lines from different organs” is too audacious, since the authors only tested it on MDCK and A549 cell lines! The authors should refrain a bit their enthusiasm or perform the experiment on several human cell lines from different organs that can be targeted by influenza virus (heart, spleen, brain, liver, lymphoid ograns).

Comments on the Quality of English Language

As mentioned in my comments and suggestions to the authors, i noticed some spelling and typing errors. Furthermore, several sentences written in the introduction section are out of place and should be either in the results or in the discussion section.

Author Response

Dr. Nico Jehmlich

Editor-in-Chief

Microorganisms

Mar 24, 2024

Dear Nico Jehmlich:

We are grateful for the positive and constructive feedback we received from you and the reviewers on our manuscript titled "A noble extract of Pseudomonas sp. M20A4R8 efficiently controlling the influenza virus-induced cell death," revised in microorganisms-2924133. We have taken into account all the suggestions made by the reviewers and have submitted a modified manuscript with additional results to address their concerns. In the revised manuscript, we have indicated the extra information or modifications made in response to the issues raised by the reviewers in yellow (first reviewer) and green (second reviewer) highlights, respectively.

We have attached a point-by-point response to each reviewer's comments along with the manuscript. Please do not hesitate to contact us if you need any additional information. We appreciate your consideration and look forward to hearing from you soon.

Sincerely yours

Jun-Gyu Park, DVM, PhD

Assistant Professor

Department of Veterinary Zoonotic Diseases

College of Veterinary Medicine

Chonnam National University

77 Yongbong-ro, Buk-gu, Gwangju 61186,

Republic of Korea

Office: +82-62-530-2845

Sang-Ik Park, DVM, PhD

Professor

Department of Veterinary Pathology

College of Veterinary Medicine and BK21 FOUR Program

Chonnam National University

77 Yongbong-ro, Buk-gu, Gwangju 61186,

Republic of Korea

Office: +82-62-530-2843

For research article

Response to Reviewer 1’s Comments

  1. Summary

We would like to express our sincere gratitude for your constructive feedback on our manuscript. We have carefully considered all of your suggestions and have made corrections and modifications to address each point raised. In the revised manuscript, any additional information or changes made in response to the concerns raised by reviewer 1 have been highlighted in yellow highlight for easy reference

2. Questions for General Evaluation

Reviewer’s Evaluation

Does the introduction provide sufficient background and include all relevant references?

Yes

Are all the cited references relevant to the research?

Yes

Is the research design appropriate?

Yes

Are the methods adequately described?

Yes

Are the results clearly presented?

Can be improved

Are the conclusions supported by the results?

Yes

  1. Point-by-point response to Comments and Suggestions for Authors

Comments 1: Some sentences should in fact be either in the results or even in the discussion part : sentence lines 67-70; sentences lines 71-78; sentences lines 83-90.

Response 1: Thank you for pointing this out. We agree with your observations that some sentences should be included in either the Results or Discussion section. Since the Result and Discussion sections already contain this information, we have removed it from the introduction to prevent redundancy. Instead, we addressed antiviral mechanisms more, specifically the interplay between influenza infection and apoptosis, including autophagy [lines 68-90].

Comments 2: The authors could detail a bit the interplay between influenza infection and apoptosis. It is not clear like it is written.

Response 2: Thank you for drawing my attention to this matter. As stated above, I have provided further elucidation on the intricate molecular interplay between influenza infection and programmed cell death, including autophagy [lines 68-90].

Comments 3: Line 218 : coronavirus ? not necessary to mention in this article.

Response 3: We apologize for this error. We removed it from the revised manuscript [line 219].

Comments 4: Lines 246 and 260 : mention which primary antibody has been used : what is the antigen and give references of these antibodies.

Response 4: We appreciate your attention to our work. We have taken note of your feedback and have made the necessary corrections in the revised manuscript. With regards to the primary antibody, we utilized the same antibody from Abcam (UK, Cambridge) as outlined in the protocol by Baek et al. (2022), [lines 151-152 and 248-249].

Additionally, we have provided extra information about a kit and chemicals as follows:

  • marine agar 2216 (MA; Difco) [original manuscript, line: 108] à marine agar 2216 (Difco, Franklin Lakes, NJ, USA) [revised manuscript, line 108]
  • dissolved in dimethyl sulfoxide (DMSO) [line: 143] à dissolved in dimethyl sulfoxide (DMSO) (Thermo Fisher Scientific, Waltham, MA, USA) [lines 143-144]
  • diluted 10 times to DMEM [line: 159] à diluted 10 times to DMEM (Welgene, Namcheon, KR) [line 160]
  • (Sigma-Aldrich) as previously reported [line: 189] à (Sigma-Aldrich, St. Louis, MA, USA) as previously reported) [line 190]
  • a 5% crystal violet solution (DUKSAN) [lines: 237-238] à 5% crystal violet solution (DUKSAN, Busan, KR) [lines 238-239]
  • to determine the protein of viral progeny [line: 240] à to determine the protein of viral progeny using an in situ cell death detection kit (Roche, Basel, CH) [lines 242-243]
  • DAPI was used to stain the nucleus [line: 248] à DAPI (Merck Millipore, Burlington, MS, USA) was used to stain the nucleus [lines 251-252]

Comments 5: Lines 247 and 261 : as well, give the references of the secondary antibodies.

Response 5: We would like to express our gratitude for your kind attention. We would like to inform you that we have incorporated the relevant information pertaining to the secondary antibody in our manuscript antibody [lines 250-251 and lines 271-272].

Comments 6: Please indicate the cell line used in figures 2, S1, 3, 4, 5A-D.

Response 6: Thank you for drawing our attention to the matter. We would like to inform you that we have incorporated the relevant information in the revised manuscript. The updated manuscript now includes the pertinent details in figures 2 [line 322], S1 [line 512], 3 [line 337], 4 [line 407], and 5A-D [line 432, 433].

Comments 7: Sentence line 327-328 : The viral inoculum (at an MOI of 0.1) was mixed with the extract and absorbed into the MDCK cell, add “for one hour incubation”

Response 7: Thank you for your valuable counsel. We have added the information in the revised manuscript [line 348].

Comments 8: Figure 4A/B : the authors should present the flow cytometry results on mock-infected cells in order to assess that the bacterial extract is not inducing itself apoptosis, at least as supplementary data.

Response 8: Thank you for drawing our attention to this pertinent matter. We have taken your feedback into consideration and have made appropriate revisions in the form of including the mock-infected, vehicle-treated group and mock-infected, M20A4R8 extract-treated group in the revised Figure 4 [line 406], describing the figure legend accordingly [lines 407-415]. Furthermore, we have addressed the result again in the revised manuscript [lines 382-384].

Comments 9: Sentences lines 394-404: the conclusion that “M20A4R8 extract has in vitro broad-spectrum antiviral potential, including multiple influenza viruses in various cell lines from different organs” is too audacious, since the authors only tested it on MDCK and A549 cell lines! The authors should refrain a bit their enthusiasm or perform the experiment on several human cell lines from different organs that can be targeted by influenza virus (heart, spleen, brain, liver, lymphoid organs).

Response 9: We thank the reviewer for the important comment and agree that it is too outreach, given that the experiment was only performed in 2 cell lines. So, we toned down the result in the revised manuscript [lines 427-428].  

  1. Response to Comments on the Quality of English Language

Point 1: As mentioned in my comments and suggestions to the authors, i noticed some spelling and typing errors. Furthermore, several sentences written in the introduction section are out of place and should be either in the results or in the discussion section.

Response 1: Thank you for your insightful comments and constructive feedback. We have thoroughly reviewed the manuscript and have taken the necessary steps to address the issues you raised. Specifically, we have meticulously checked for and corrected spelling and typing errors. Additionally, we have reorganized the introduction section to ensure all information is correctly placed in their respective sections and removed any redundant information. The corrections made are as follows, except the above issues already addressed:

  • Phylogeny Analysis [original manuscript, line 114]à Phylogenetic Analysis [revised manuscript, line 114]
  • Pseudomonas and Kineobactrum species [lines 134-135] à Pseudomonas, and Kineobactrum species [lines 134-135]
  • The half cytotoxic concentration (CC50) and IC50 of the Pseudomonas M20A4R8 ex-tract was calculated after the generation of the dose-response curve [lines 196-197] à The half-cytotoxic concentration (CC50) and IC50 of the M20A4R8 extract were calculated after generating a dose-response curve [lines 197-198]
  • an MOI 0.1 [line 203] à at an MOI of 0.1 [lines 204-205]
  • primer pairs; [line 222] à primer pairs: [line 223]
  • with 5% M [line 234] à with 5% CO2 [line 235]
  • M20A4R8 strain [line 274] à the M20A4R8 strain [line 291]
  • M20A4R8 strain [line 276] à the M20A4R8 strain [lines 293-294]
  • Paraglaciecola aquimarina and Kineobactrum salinum [lines 134-135, 289-290] à Paraglaciecola aquimarina, and Kineobactrum salinum [lines 134-135, 315-316]
  • relationships of M20A4R8 strain [line 296] à relationships of the M20A4R8 strain [line 299]
  • agains [lien: 303] à against [line 321]
  • IAV HA [line 348] à IAV hemagglutinin (HA) [line 368]
  • 3 independent experiments [lines 486-487] à 3 independent experiments. N.D., Not determined [line 517]
  • before binding of the virus [line 489] à before the viral binding [line 519]
  • binding of A/PR8 strain [line 490]à binding of the A/PR8 strain [line 520]
  • was measure [line 491] à were measured [lines 521-522]

Reviewer 2 Report

Comments and Suggestions for Authors

In the current manuscript, Jung et al investigated the anti-influenza virus activity of an extract prepared from the novel marine bacterium Pseudomonas sp. M20A4R8. Using an in vitro infection model, the authors show that M20A4R8 extract showed low cytotoxicity in uninfected MDCK cells and inhibited virus-induced CPE in influenza A/PR8-infected cells in a dose-dependent manner. Further, experimental data suggested that the M20A4R8 extract affected the later stages of influenza A virus infection (genome replication, particle formation) rather than viral attachment and entry. Data obtained using a flow cytometric assay for DNA fragmentation suggest that the M20A4R8 might counteract apoptotic death induced by influenza A virus infection. Lastly, the authors show that M20A4R8 exhibits a broad-spectrum anti-viral effect against influenza A (H3N2) and influenza B strains of virus.

Though the rationale for the study and study design mirror quite closely with a recent manuscript published by the authors (PMID# 38326489), the findings in the current manuscript differ enough to suggest that the marine extracts tested in each study likely possess different anti-viral properties. The authors sufficiently describe the methodologies used and the results are presented in a clear and understandable format. The authors' conclusion are, in general, supported by their experimental data

There are several issues with the current manuscript that need to be addressed before publication is recommended:

1. For Figure 2, it is unclear what the symbols represent (means? medians?). Also, the number of replicates and independent experiments needs to be added.

2. line 339-340: since only single representative IFA images are shown the language used in data interpretation must be softened (i.e., data suggest rather than show). The data are not strong enough to support that M20A4R8 extract inhibits viral protein synthesis.

3. Related, it would be more convincing if the authors used a quantitative approach rather than qualitative approach to assess the impact of M20A4R8 treatment on viral protein synthesis (e.g., fluorescence intensity measurements from multiple IFA images, Western Blot for NP protein).

4. Figure 4: the authors need to show their gating strategy (FSC vs SSC). Additionally, a panel showing data from uninfected, vehicle treated cells is needed to establish that the quadrant gates are correctly positioned.

5. Figure 5: the authors need to explain what the symbols represent, as well as the number of replicates and independent experiments.

6. line 445: data is not strong enough to indicate whether M20A4R8 inhibits viral protein synthesis. Inhibition of viral packaging/assembly is also a strong possibility.

Author Response

Dr. Nico Jehmlich

Editor-in-Chief

Microorganisms

Mar 24, 2024

Dear Nico Jehmlich:

We are grateful for the positive and constructive feedback we received from you and the reviewers on our manuscript titled "A noble extract of Pseudomonas sp. M20A4R8 efficiently controlling the influenza virus-induced cell death," revised in microorganisms-2924133. We have taken into account all the suggestions made by the reviewers and have submitted a modified manuscript with additional results to address their concerns. In the revised manuscript, we have indicated the extra information or modifications made in response to the issues raised by the reviewers in yellow (first reviewer) and green (second reviewer) highlights, respectively.

We have attached a point-by-point response to each reviewer's comments along with the manuscript. Please do not hesitate to contact us if you need any additional information. We appreciate your consideration and look forward to hearing from you soon.

Sincerely yours

Jun-Gyu Park, DVM, PhD

Assistant Professor

Department of Veterinary Zoonotic Diseases

College of Veterinary Medicine

Chonnam National University

77 Yongbong-ro, Buk-gu, Gwangju 61186,

Republic of Korea

Office: +82-62-530-2845

Sang-Ik Park, DVM, PhD

Professor

Department of Veterinary Pathology

College of Veterinary Medicine and BK21 FOUR Program

Chonnam National University

77 Yongbong-ro, Buk-gu, Gwangju 61186,

Republic of Korea

Office: +82-62-530-2843

For research article

Response to Reviewer 2’s Comments

  1. Summary

We would like to express our gratitude for the valuable feedback you provided on our manuscript. We have carefully reviewed all of your suggestions and made the necessary corrections and modifications to address each point you raised. In the revised version of the manuscript, we have highlighted any additional information or changes made in response to reviewer 2's concerns in green for easy reference. Thank you again for your help in improving our manuscript.

2. Questions for General Evaluation

Reviewer’s Evaluation

Does the introduction provide sufficient background and include all relevant references?

Yes

Are all the cited references relevant to the research?

Yes

Is the research design appropriate?

Yes

Are the methods adequately described?

Yes

Are the results clearly presented?

Can be improved

Are the conclusions supported by the results?

Yes

  1. Point-by-point response to Comments and Suggestions for Authors

Comments 1: For Figure 2, it is unclear what the symbols represent (means? medians?). Also, the number of replicates and independent experiments needs to be added.

Response 1: Thank you for pointing out this important issue. We have included the information in the revised manuscript [lines 324-325].

Comments 2: line 339-340: since only single representative IFA images are shown the language used in data interpretation must be softened (i.e., data suggest rather than show). The data are not strong enough to support that M20A4R8 extract inhibits viral protein synthesis.

Response 2: Thank you for bringing this critical matter to our attention. We have taken the necessary steps to address the issue by refining the language and providing a quantification result for IFA in the revised Figure 3, thereby enhancing the reliability of the inhibitory effect [lines 338-341 and lines 343-344]. Furthermore, we have incorporated the details pertaining to the quantification methodology in the revised manuscript [lines 254-259].

Comments 3: Related, it would be more convincing if the authors used a quantitative approach rather than qualitative approach to assess the impact of M20A4R8 treatment on viral protein synthesis (e.g., fluorescence intensity measurements from multiple IFA images, Western Blot for NP protein).

Response 3: Thank you for bringing this critical matter to our attention. We have taken your feedback into consideration and addressed the issue pertaining to the quantification approach using IFA images, as mentioned above. We greatly appreciate your valuable input, which has undoubtedly contributed to the quality and reliability of our manuscript.

Regarding the feasibility of employing Western Blot to prove the inhibitory effect on viral protein synthesis, we regret to inform you that we were unable to proceed with the experiment due to time and resource constraints. However, we have taken note of your recommendation and included the relevant information as a potential limitation in the Discussion section [lines 497-509].

Comments 4: Figure 4: the authors need to show their gating strategy (FSC vs SSC). Additionally, a panel showing data from uninfected, vehicle treated cells is needed to establish that the quadrant gates are correctly positioned.

Response 4: We appreciate your attention to this crucial matter. As per the suggestion of another reviewer, we have taken the necessary steps to address the concerns raised and have included the results related to the mock-infected, vehicle-treated group and the mock-infected, M20A4R8 extract-treated group in the revised Figure 4 [lines 407-415] and manuscript [lines 382-384] in yellow highlight.

We would like to assure you that we have included a detailed methodology in the revised manuscript [lines 276-281] for the gating strategy. We did not selectively choose a particular population for this analysis. Instead, we included all possible cells involved in the late-stage apoptosis or viral infection to evaluate the general virus-induced apoptosis effect.

Comments 5: Figure 5: the authors need to explain what the symbols represent, as well as the number of replicates and independent experiments.

Response 5: Thank you for pointing out this important issue. We have included the information in the revised manuscript [lines 434-435].

Comments 5: line 445: data is not strong enough to indicate whether M20A4R8 inhibits viral protein synthesis. Inhibition of viral packaging/assembly is also a strong possibility.

Response 5: Thank you for bringing this crucial matter to our attention. We value your input and have diligently incorporated the relevant details into the revised manuscript. Additionally, we have taken great care to reconstruct the paragraph [lines 497-509] to ensure the accuracy and validity of the content presented.

Round 2

Reviewer 2 Report

Comments and Suggestions for Authors

I thank the authors' for their thorough response to my comments and suggestions. All of my major concerns have been addressed. The only minor revision I have is that the authors clarify in the Methods sections the number of replicates used in the following assays:

A. 2.5 Cytotoxicity Assessment: assays performed in 96 well plates; unclear how many wells were used for each treatment for each independent experiment. 

B. 2.10 TCID50 Assay: assays performed in 96 well plates; unclear how many wells were used for each treatment for each independent experiment.

c. 2.12 Flow Cytometry of DNA Strand Break: assays performed in 6 well plates; unclear how many wells were used for each treatment for each independent experiment.

Author Response

Dr. Nico Jehmlich

Editor-in-Chief

Microorganisms

Mar 25, 2024

Dear Nico Jehmlich:

We are writing to express our gratitude for the continuous, constructive feedback you and the reviewers provided on our manuscript titled "A noble extract of Pseudomonas sp. M20A4R8 efficiently controlling the influenza virus-induced cell death," which we have revised and resubmitted to microorganisms-2924133. We would like to take this opportunity to thank you for the time and effort you invested in reviewing our work. The thorough review and valuable input have significantly impacted the quality of our research, and we appreciate your dedication to helping us improve.

We have carefully considered and incorporated all the changes you suggested into the revised manuscript. We have highlighted them in green to make them easier for you to identify.

Once again, we sincerely appreciate your thoughtful feedback and your commitment to advancing the field. Your input has been invaluable, and we are grateful for your contribution to our research.

Sincerely yours

Jun-Gyu Park, DVM, PhD

Assistant Professor

Department of Veterinary Zoonotic Diseases

College of Veterinary Medicine

Chonnam National University

77 Yongbong-ro, Buk-gu, Gwangju 61186,

Republic of Korea

Office: +82-62-530-2845

Sang-Ik Park, DVM, PhD

Professor

Department of Veterinary Pathology

College of Veterinary Medicine and BK21 FOUR Program

Chonnam National University

77 Yongbong-ro, Buk-gu, Gwangju 61186,

Republic of Korea

Office: +82-62-530-2843

For research article

Response to Reviewer 2’s Comments

  1. Summary

Thank you for your diligent review of our manuscript, which proved instrumental in improving it. Your unwavering commitment and efforts are deeply appreciated, and we are grateful for your valuable input. We have carefully considered your comment and made the necessary modifications to the revised manuscript, which are highlighted in green. Once again, thank you for taking the time to provide thoughtful feedback.

2. Questions for General Evaluation

Reviewer’s Evaluation

Does the introduction provide sufficient background and include all relevant references?

Yes

Are all the cited references relevant to the research?

Yes

Is the research design appropriate?

Yes

Are the methods adequately described?

Yes

Are the results clearly presented?

Yes

Are the conclusions supported by the results?

Yes

  1. Point-by-point response to Comments and Suggestions for Authors

Comments 1: 2.5 Cytotoxicity Assessment: assays performed in 96 well plates; unclear how many wells were used for each treatment for each independent experiment.

Response 1: Thank you for pointing out this important issue. We have included the information in the revised manuscript [lines 164-165].

Comments 2:  2.10 TCID50 Assay: assays performed in 96 well plates; unclear how many wells were used for each treatment for each independent experiment.

Response 2: Thank you for pointing out this important issue. We have included the information in the revised manuscript [lines 241-242].

Comments 3: 2.12 Flow Cytometry of DNA Strand Break: assays performed in 6 well plates; unclear how many wells were used for each treatment for each independent experiment.

Response 3: Thank you for pointing out this important issue. We have included the information in the revised manuscript [lines 283-284].
